# Finding Physical Adversarial Examples for Autonomous Driving with Fast and Differentiable Image Compositing

## Abstract

There is considerable evidence that deep neural networks are vulnerable to adversarial perturbations applied directly to their digital inputs. However, it remains an open question whether this translates to vulnerabilities in real-world systems. Specifically, in the context of image inputs to autonomous driving systems, an attack can be achieved only by modifying the physical environment, so as to ensure that the resulting stream of video inputs to the car's controller leads to incorrect driving decisions. Inducing this effect on the video inputs indirectly through the environment requires accounting for system dynamics and tracking viewpoint changes. We propose a scalable and efficient approach for finding adversarial physical modifications, using a differentiable approximation for the mapping from environmental modifications—namely, rectangles drawn on the road—to the corresponding video inputs to the controller network. Given the color, location, position, and orientation parameters of the rectangles, our mapping composites them onto pre-recorded video streams of the original environment. Our mapping accounts for geometric and color variations, is differentiable with respect to rectangle parameters, and uses multiple original video streams obtained by varying the driving trajectory. When combined with a neural network-based controller, our approach allows the design of adversarial modifications through end-to-end gradient-based optimization. We evaluate our approach using the Carla autonomous driving simulator, and show that it is significantly more scalable and far more effective at generating attacks than a prior black-box approach based on Bayesian Optimization.

## 1 Introduction

Computer vision has made revolutionary advances in recent years by leveraging a combination of deep neural network architectures with abundant high-quality perceptual data. One of the transformative applications of computational perception is autonomous driving, with autonomous cars and trucks already being evaluated for use in geofenced settings, and partial autonomy, such as highway assistance, leveraging state-of-the-art perception embedded in vehicles available to consumers. However, a history of tragic crashes involving autonomous driving, most notably Tesla (Thorbecke, 2020) and Uber (Hawkins, 2019) reveals that modern perceptual architectures still have some limitations even in non-adversarial driving environments. In addition, and more concerning, is the increasing abundance of evidence that state-of-the-art deep neural networks used in perception tasks are highly vulnerable to *adversarial perturbations*, or imperceptible noise that is added to an input image and deliberately designed to cause misclassification (Goodfellow et al., 2014; Yuan et al., 2019; Modas et al., 2020). Furthermore, several lines of work consider specifically *physical adversarial examples* which modify the *scene* being captured by a camera, rather than the image (Kurakin et al., 2016; Eykholt et al., 2018; Sitawarin et al., 2018; Dutta, 2018; Duan et al., 2020).

Despite this body of evidence demonstrating vulnerabilities in deep neural network perceptual architectures, it is nevertheless not evident that such vulnerabilities are consequential in realistic autonomous driving, even if primarily using cameras for perception. First, most such attacks involve independent perturbations to a given input image. Autonomous driving is a dynamical system, so that a fixed adversarial perturbation to a scene is perceived through a series of distinct, but highly interdependent perspectives. Second, self-driving is a complex system that maps perceptual inputs

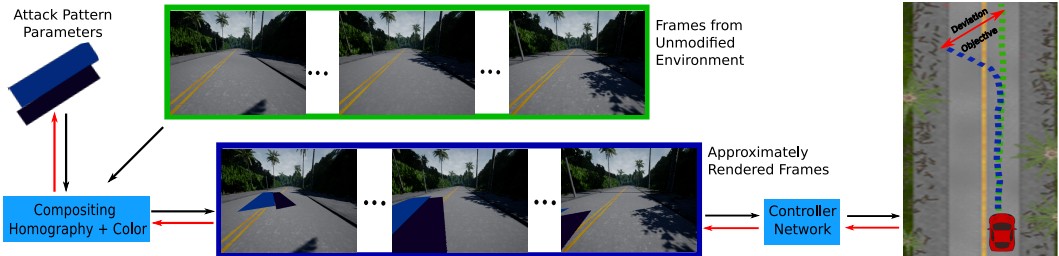

Figure 1: Overview. We collect and calibrate frames from the unmodified environment (shown in the green box), and given a choice of attack pattern parameters, composite the pattern to create approximate renderings of frames corresponding to placing the pattern in the environment. Our composition function is differentiable with respect to the attack pattern parameters, and we are thus able to use end-to-end gradient-based optimization when attacking a differentiable control network, to cause the network to output incorrect controls that cause the vehicle to deviate from its intended trajectory (from the green to the blue trajectory, as shown in the right column), and crash.

to control outputs. Consequently, even if we succeed in causing the control outputs to deviate from normal, the vehicle will now perceive a sequence of frames that is *different from those encountered on its normal path*, and typically deploy self-correcting behavior in response. For example, if the vehicle is driving straight and then begins swerving towards the opposite lane, its own perception will inform the control that it's going in the wrong direction, and the controller will steer it back on course.

To address these limitations, Bayesian Optimization (BO) (Archetti and Candelieri, 2019) was recently proposed as a way to design physical adversarial examples (2 black rectangles on road pavement) in Carla autonomous driving simulations (Dosovitskiy et al., 2017) against end-to-end autonomous driving architectures (Boloor et al., 2020). The key challenge with this approach, however, is that attack design must execute actual experiments (e.g., simulations, or actual driving) for a larger number of iterations (1000 in the work above), making it impractical for large-scale or physical driving evaluation. Furthermore, it is not clear how well BO scales as we increase the complexity of the adversarial space beyond 2 black rectangles.

We propose a highly scalable framework for designing physically realizable adversarial examples against end-to-end autonomous driving architectures. Our framework is illustrated in Figure 1, and develops a differentiable pipeline for digitally approximating driving scenarios. The proposed approximation makes use of image compositing, learning homography and color mappings from a birds-eye view of embedded adversarial examples to projections of these in images based on actual driving frames, and sampling sequences of actual frames with small added random noise to control to ensure adequate sampling of possible perspectives. The entire process can then be fed into automatic differentiators to obtain adversarial examples that maximize a car's deviation from its normal sequence of controls (e.g., steering angle) for a target driving scenario.

We evaluate the proposed framework using Carla simulations in comparison with the state-of-the-art BO method. Our experiments show that the resulting attacks are significantly stronger, with effects on induced deviations and road infractions often considerably outperforming BO, at a small fraction of actual driving runs required for training. Furthermore, we show that our approach yields attacks that are robust to unforeseen variations in weather and visibility.

**Related Work:** Attacks on deep neural networks for computer vision tasks has been a subject of extensive prior research (Goodfellow et al., 2014; Yuan et al., 2019; Modas et al., 2020; Vorobeychik and Kantarcioglu, 2018). The most common variation introduces imperceptible noise to pixels of an image in order to induce error in predictions, such as misclassification of the image or failure to detect an object in it. A more recent line of research has investigated *physical adversarial examples* (Kurakin et al., 2016; Athalye et al., 2017; Eykholt et al., 2018; Sitawarin et al., 2018; Dutta, 2018; Duan et al., 2020), where the explicit goal is to implement these in the physical scene, so that the images of the scene subsequently captured by the camera and fed into a deep neural network result in a prediction error. In a related effort, Liu et al. (2019) developed a differentiable renderer that allows the attacker to devise higher-level perturbations of an image scene, such as geometry and lighting, through a differentiable renderer. However, most of these approaches attack a fixed input

scene, whereas autonomous driving is a complex dynamical system. Several recent approaches investigate physical attacks on autonomous driving that attempt to account for the fact that a single object is modified and viewed through a series of frames (Ackerman, 2019; Kong et al., 2020; Boloor et al., 2020). However, these either still consider digital attacks, albeit restricted to a small area (e.g., replacing a road sign with a noisy road sign) and do not consider a vehicles self-correcting behavior (for example, (Kong et al., 2020)), or rely on many expensive driving experiments in order to identify adversarial patterns (Boloor et al., 2020).

## 2 PROPOSED METHOD

Autonomous driving systems are equipped with decision algorithms that produce control signals for a vehicle, based on high-level instructions—such as a given route or destination—and inputs from cameras and other sensors that make continuous measurements of the vehicle's physical environment. We assume the decision algorithm is in the form of a differentiable function—such as a neural network—that maps video frames from the camera, along with other inputs, to the control outputs. Given such a network or function, our goal is to determine if it is vulnerable to attack. Specifically, we seek to build a scalable and efficient method to find modifications that can be applied to the physical environment, and result in a stream of video frames which cause the control network to produce output signals that disrupt the vehicle's operation, moving it away from the expected ideal trajectory.

This task is challenging since the relationship between modifications to the physical environment and the network's inputs is complex: the video frames correspond to images of the environment from a sequence of changing viewpoints, where the sequence itself depends on the network's control outputs. The precise effect of any given modification can be determined only by actually driving the vehicle in the modified environment, or by using a high-quality simulator with a sophisticated physics engine. However, it is expensive to use actual driving or an expensive simulator when searching for the right modification, since neither process is differentiable with respect to parameters of the modification, and would require repeated trials with candidate modifications in every step of the search process.

Instead, we propose a fast approximation to produce video frames for a given environment given a candidate modification that is differentiable with respect to parameters of the modification. Our approach requires a small number of initial calibration runs—of actual driving or a sophisticated simulator—after which, the search for optimal parameters can be carried out with end-to-end gradient-based optimization. Specifically, we consider the case when the modification takes the form of figures (such as rectangles) drawn on a restricted stretch of the road, and task the optimization with finding their optimal shape and color so as to maximize deviation from the controller's trajectory prior to modification. We now describe our model for the physical modification, our approximate mapping to create video frames for a given modification, and our optimization approach based on this mapping.

### 2.1 PARAMETERIZED PHYSICAL MODIFICATION

We assume modifications are in the form of a collection of $K$ figures (e.g., rectangles) that will be painted on a flat surface in the environment (e.g., road). Let $\Phi = \{x_k^S, x_k^C\}_{k=1}^K$ denote the parameters of this modification, with $x_k^S$ corresponding to the shape parameters, and $x_k^C$ the RGB color, of the $k^{th}$ figure. These parameters are defined with respect to co-ordinates in some canonical—say, top-down—view of the surface.

We let $M(n_c; x^S)$ denote a scalar-valued mask that represents whether a pixel at spatial location $n_c \in \mathbb{Z}^2$ in the canonical view is within a figure with shape parameters $x^S$. This function depends simply on the chosen geometry of the figures, and has value of $1$ for pixels within the figure, $0$ for those outside, and real values between $0$ and $1$ for those near the boundary (representing partial occupancy on a discrete pixel grid to prevent aliasing artifacts).

Since the spatial extents for different figures may overlap, we next account for occlusions by assuming that the lines will be painted in order. Accordingly, we define a series of visibility functions $V_k(n_c; \Phi)$, each representing the visibility of the $k^{th}$ figure at pixel $n_c$, after accounting for occlusions. We set the function for the last figure as $V_K(n_c; \Phi) = M(n_c; x_K^S)$, and for the other figures with $k < K$,

$$V_k(n_c; \Phi) = M(n_c; x_k^S) \prod_{k'=k+1}^{K} (1 - V_{k'}(n_c; \Phi)). \tag{1}$$

## 2.2 APPROXIMATE FRAMES VIA COMPOSITING

The next step in our pipeline deals with generating the video inputs that the controller network is expected to receive from a modified environment for given parameter values $\Phi$. These frames will represent views of the environment, including the surface with the painted figures, from a sequence of viewpoints as the car drives through the scene. Of course, the precise viewpoint sequence will depend on the trajectory of the car, which will depend on the control outputs from the network, that in turn depends on the frames. Instead of modeling the precise trajectory for every modification, we instead consider a small set of $T$ representative trajectories, corresponding to those that the vehicle will follow when driven with small perturbations around control signals the network outputs, when operating in the unmodified environment. In our experiments, we use $T = 4$ trajectories: one from driving the car with the actual output control signals, and three from random noise added to these outputs. Given the fact that actual control is closed-loop, it is not evident that this simple approach would work; however, our experiments below show that it is remarkably effective, despite its simplicity.

This gives $T$ sequences of video frames, one for each trajectory, where we assume each sequence contains $F$ frames. We let $\tilde{I}_f^t(n)$ denote the $f^{th}$ "clean" image in the $t^{th}$ sequence, representing a view of the environment without any modifications. Here, $n \in \mathbb{Z}^2$ indexes pixel location within each image, and the intensity vector $\tilde{I}_f^t(n) \in \mathbb{R}^3$ at each location corresponding to the recorded RGB values. These clean images can be obtained by driving the car—actually, or in simulation—in the original environment.

For each frame in each sequence, we also determine a spatial mapping $n_c = G_f^t(n)$ that maps pixel locations in the image to the canonical view. We model each $G_f^t(n)$ as a homography: the parameters of which can be determined by either using correspondences between each image and the canonical view of the surface—from calibration patterns rendered using the simulator, or from user input—or by calibrating the vehicle's camera and making accurate measurements of ego-motion when the vehicle is being driven. Additionally, we also determine color mapping parameters $C_f^t \in \mathbb{R}^{3 \times 3}, b_f^t \in \mathbb{R}^3$ for each frame representing an approximate linear relationship between the RGB colors $x^C$ of the painted figures, and their colors as visible in each frame. These parameters are also determined through calibration. Given this set of clean frames and the geometric and color mapping parameters, we generate corresponding frames with views of the modified environment simply as:

$$I_f^t(n; \Phi) = \left(1 - \sum_{k=1}^{K} V_k(G_f^t(n); \Phi)\right) \tilde{I}_f^t(n) + \sum_{k=1}^{K} V_k(G_f^t(n); \Phi) \left(C_f^t x_k^C + b_f^t\right). \quad (2)$$

## 2.3 GRADIENT-BASED OPTIMIZATION

Given the "forward" process of generating a set of frames for a given set of modification values $\Phi$, we finally describe our approach to finding the value of $\Phi$ that misleads the control network. We let $D(\{I_f[n]\}_f)$ denote the controller network, which takes as input a sequence of frames $\{I_f[n]\}$ in a single trajectory and generates a corresponding sequence of real-valued control signals, such as a steering angle at each time instant. Our goal is to find the value of $\Phi$ that maximizes deviation of these outputs from those for an unmodified environment. We cast this as minimization of a loss over our $T$ trajectories, i.e.,

$$\Phi = \arg\min - \sum_{t=1}^{T} \delta\left(D(\{I_f^t[n, \Phi]\}_f), D(\{\tilde{I}_f^t[n]\}_f)\right), \quad (3)$$

in which $\delta(\cdot, \cdot)$ measures divergence between two sequences of control outputs. In our experiments where the control network outputs a sequence of steering angles, we use the absolute value of the sum of the differences between the angles as our divergence metric.

We carry out this optimization iteratively using gradient descent with Adam (Kingma and Ba, 2014) because, as shown next, we are able to compute gradients of the loss in (3) with respect to the modification parameters $\Phi$. Since the controller network $D(\cdot)$ is assumed to be a neural network (or a differentiable function), we are able to use standard back-propagation to compute gradients $\nabla\left(I_f^t(n; \Phi)\right)$ of the loss with respect to each intensity in the images of the modified environment.

The gradients with respect to the color parameters $\{x_k^C\}$ can then be computed based on (2) as:

$$\nabla(x_k^C) = \sum_{t,f} \left(C_f^t\right)^T \left(\sum_n V_k(G_f^t(n); \Phi) \, \nabla\left(I_f^t(n; \Phi)\right)\right). \tag{4}$$

Computing gradients with respect to the shape parameters $\{x_k^S\}$ requires an approximation, since the mask functions $M(\cdot)$ are not generally differentiable with respect to these parameters. We adopt a simple local linearization approach: for every scalar parameter $\theta$ in the shape parameters $x_k^S$ for each figure, we discretize its range into a fixed set of equally separated values. Then, given the current (continuous) value of $\theta$, we let $\theta^-$ and $\theta^+$ represent the consecutive pair of values in this discrete set, such that $\theta^- \le \theta \le \theta^+$, and denote $\Phi_{\theta-}$ and $\Phi_{\theta+}$ the full set of current parameter values, with $\theta$ replaced by $\theta^+$ and $\theta^-$ respectively. We make the approximation that if $\alpha \in \mathbb{R}$ such that $\theta = \alpha\theta^+ + (1 - \alpha)\theta^-$, then

$$I_f^t(n; \Phi) \approx \alpha I_f^t(n; \Phi_{\theta+}) + (1 - \alpha) I_f^t(n; \Phi_{\theta-}). \tag{5}$$

Therefore, although we only use frames $I_f^t(n; \Phi)$ with the actual value of $\Phi$ as input to the control network, we also generate an extra pair of images $I_f^t(n; \Phi_{\theta-}), I_f^t(n; \Phi_{\theta+})$ for each frame for every element $\theta$ of the shape parameters. We then use these frames to compute parameter gradients as:

$$\nabla(\alpha) = \sum_{t,f,n} \nabla\left(I_f^t(n; \Phi)\right)\left(I_f^t(n; \Phi_{\theta+}) - I_f^t(n; \Phi_{\theta-})\right), \quad \nabla(\theta) = \nabla(\alpha) \times (\theta^+ - \theta^-). \tag{6}$$

Therefore, our approximate framework allows us to compute gradients of the loss in (3) with respect to all the shape and color parameters in $\Phi$, and use gradient descent to minimize this loss.

## 3 EXPERIMENTS

In this section, we experimentally evaluate our approach, which we refer to as *GradOpt*, and demonstrate that it is both more efficient—requiring fewer actual or simulated drives—and effective—in successfully finding attack patterns—than the state-of-the-art Bayesian Optimization (*BO*) method (Boloor et al., 2020). To be able to carry out a large scale evaluation, we perform our experiments in simulation using a packaged version of the Carla simulator (Dosovitskiy et al., 2017) provided by Boloor et al. (2020) that allows the addition of painted road patterns to existing environments. Our experiments evaluate attacks against the control network that is included with Carla: this is a neural network-based controller that uses only camera inputs and outputs steering angles, and was trained using imitation learning. We run evaluations only on scenarios where this controller drives successfully without infractions in the unmodified environment.

Our experiments consider forty scenarios, each requiring driving through a stretch of road in a virtual town. Each scenario begins an episode with the vehicle spawned at a given starting waypoint, and the controller is then tasked with reaching a defined destination waypoint. The episode runs until the vehicle reaches this destination or a time-limit expires (e.g., if the car crashes). Our scenarios are of three types: (a) the expected behavior is for the car to go *straight* (16 scenarios), (b) veer *left* (12 scenarios), or (c) *right* (12 scenarios)—either through an intersection, or on a straight or curved road.

Our threat model assumes an attacker can draw a pattern on the road with the intent of causing the controller to output erroneous steering angles and the car to deviate from its intended path. Accordingly, we task both BO and our method, GradOpt, with finding such an attack pattern for each scenario. We consider patterns that are unions of rectangles (i.e., each "figure" in Sec. 2.1 is a rectangle), where the shape of each rectangle is determined by four parameters (i.e., $x_k^C \in \mathbb{R}^4$): rotation, width, length, and horizontal offset[1]. We report results when searching over shape and color parameters for different numbers of rectangles $K$—ranging from $K = 1$ (7 parameters) to $K = 5$ (35 parameters), and additionally for the single rectangle case, also when searching over only shape parameters while keeping color fixed to black (4 parameters). We learn these parameters with respect to the top-view co-ordinate frame of a canvas, that during evaluation will be passed to the simulator to be superimposed on the road (and then captured by the camera as part of the scene in all frames).

---

[1]We assume the rectangle is vertically centered prior to rotation. This is similar to the parameterization used in Boloor et al. (2020), except that we also search over length (for both GradOpt and BO).

We train both BO and GradOpt in a simulated setting without any pedestrians or other cars (this models a realistic scenario where an attacker is unlikely to test or calibrate their attacks with others present). We evaluate the success of the attack on actual simulations with Carla, in terms of two metrics. The first measures deviation between the paths with and without the attack pattern, both without pedestrians or other cars. Here, deviation is defined as:

$$\text{Deviation} = \frac{1}{2T} \sum_{t=1}^{t=T} \min_{t'} |\widetilde{W}_t - W_{t'}| + \min_{t'} |W_t - \widetilde{W}_{t'}|, \tag{7}$$

where $\widetilde{W}_t$ and $W_t$ are a sequence of vehicle locations when driving with and without the attack pattern, at a fixed set of time instances defined as those when the region of the road where the attack pattern is to be placed is visible while driving in the unmodified environment (i.e., without the pattern). The second metric in our evaluation is the total infraction penalty when driving *with* pedestrians and other cars—here we use the standard penalties as defined by the Carla Autonomous Driving Challenge (Ros et al., 2019) (where, for example, a lane violation carries a penalty of 2 points, hitting a static object or another vehicle of 6, hitting a pedestrian of 9, etc.). For each attack and scenario, we run simulations ten times (with the simulator randomly spawning pedestrians and cars each time) and average the infraction penalty scores. As mentioned before, the controller has zero infractions when driving in the unmodified environment in all these scenarios. Finally, while both BO and GradOpt are trained in a *clear noon* weather setting, we measure infractions on that setting as well as three others: *cloudy noon*, *rainy noon*, and *clear sunset*.

## 3.1 ATTACK OPTIMIZATION

**Proposed Method:** Our approach requires two steps for every scenario: (1) collecting a set of frame sequences and calibrating them, and (2) performing gradient-based optimization. For (1), we collect frames from four trajectories—one obtained by driving the car with the direct outputs of the controller, and three others with perturbations to these outputs. We estimate the homographies $G_f^t$ for every frame $f$ in trajectory $t$ by running additional simulations with calibration patterns painted on the canvas: we run twelve simulations for each trajectory, with five calibration points of different colors in each simulation, and use the sixty automatically detected correspondences to estimate the homography matrix for every frame. We also learn a common set of color transform parameters for all frames in all trajectories, which we obtain by running the simulator twenty-two times, each time with the entire canvas painted a different high-contrast color, on the unperturbed trajectory. Therefore, our method calls the simulator a total of *74* times. Note, this is because we chose to use automated calibration to be consistent with the access provided to BO below (Boloor et al., 2020), and the number of calls could be reduced with manual annotation (or by using calibrated acceleration information from the simulator or vehicle).

Once we have a set of calibrated frames, we employ gradient-based optimization (see Sec. 2.3). We use the absolute value of the sum of differences in steering angles as the divergence metric $\delta(\cdot)$ in (3). We run the optimization with four different random starts, each for 250 iterations, for a total of 1000 iterations. We begin with a learning rate of 0.1 for each start, and drop it by a factor of 0.1 after the first 100, and the first 200 iterations. Each time we drop the learning rate, we also reset the parameters to those that yielded the lowest value of the loss thus far.

**Bayesian Optimization:** We employ BO with the same objective as ours based on the same divergence metric, and closely follow the setting in (Boloor et al., 2020)—i.e., we run the optimization for a total of 1000 iterations, of which the first 400 are random exploration. Note that while this is the same number of iterations as we use for our method, every iteration of BO requires running a full episode with the Carla simulator.

**Run-time:** Every iteration of gradient-based updates using our method takes 8 seconds, whereas simulating a full episode using the Carla simulator takes more than 80 seconds—both on an NVIDIA 1080Ti GPU. Since BO requires 1000 calls to the simulator, it takes more than **22 hours** to find the optimal attack pattern for each scenario. In contrast, we only call the simulator 74 times and then run 1000 iterations of gradient-based optimization. Ignoring the potential of parallelizing the iterations for the four random starts, this corresponds to a running time of around **four hours** per scenario. Thus, our method affords a significant advantage in computational cost, and as we show next, is also more successful at finding optimal attack patterns.

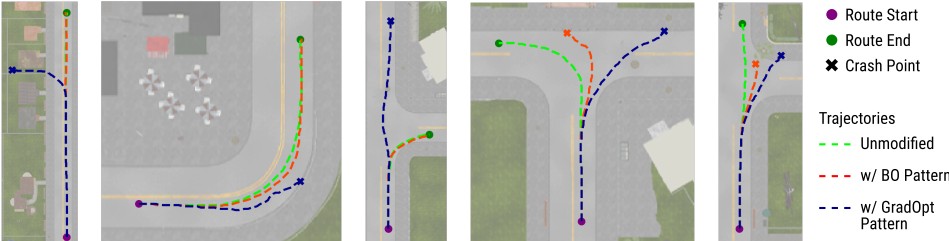

Figure 2: Trajectory deviations induced by attack. For four example scenarios, we illustrate the effect of the attack patterns found by the proposed GradOpt method and BO, by showing trajectories taken by the vehicle in the simulator when these patterns are placed on the road, and comparing these to the original trajectory in the unmodified environment.

| K (#Rect.) | Deviation | | | | Infraction Penalty | | |
|---|---|---|---|---|---|---|---|
| | BO | GradOpt | $\% \geq$ | GradOpt (T=1) | BO | GradOpt | $\% \geq$ |
| 1-b | 0.79 | *0.98* | *72%* | 0.73 | 3.84 | *3.97* | *88%* |
| 1 | 0.86 | *1.00* | *72%* | 0.82 | 4.30 | *5.21* | *75%* |
| 2 | 0.85 | *1.16* | *82%* | 0.89 | 4.50 | *6.40* | *80%* |
| 3 | 0.90 | *1.23* | *75%* | 0.95 | 4.53 | *6.03* | *88%* |
| 4 | 0.74 | *1.29* | *80%* | 1.10 | 3.69 | *6.40* | *90%* |
| 5 | 0.82 | *1.33* | *82%* | 0.99 | 4.70 | *7.43* | *82%* |

Table 1: Average deviation (in simulations without cars or pedestrians) and infraction penalties (averaged across 10 simulations with cars and pedestrians) across all scenarios for GradOpt (proposed) and BO, when searching for parameters of different numbers of rectangles (1-b refers to searching only the shape parameters of one rectangle, fixing its color to black). All simulations are in "clear noon" weather as also used for attack optimization. The $\% \geq$ column reports the percentage of instances where GradOpt has higher or equal score than BO. Also shown are average deviations when using GradOpt with only one trajectory ($T = 1$), instead of four, for optimization.

## 3.2    RESULTS

We begin by describing an evaluation of the proposed GradOpt method's ability to find successful attacks in all forty scenarios, and comparing this to the prior BO approach of Boloor et al. (2020). Table 1 reports quantitative results in terms of the metrics deviation and infraction penalty discussed above, computed with simulations in the same standard weather setting as used for attack optimization. We report averages for these metrics over all scenarios, as well as the percentage of cases when GradOpt performs as well or better than BO, when searching for patterns of increasing complexity (in terms of number of rectangles $K$). We find that GradOpt is significantly more successful than BO, despite also being computationally less expensive as discussed earlier. It has higher average values of both deviation and infraction penalties, with the gap growing significantly for higher values of $K$—indicating that GradOpt is much better able to carry out optimization in a higher-dimensional parameter space and leverage the ability to use more complex patterns. Moreover, we find that it yields attack patterns that as or more successful than BO in more than 70% of cases in all settings, with the advantage rising to 82% for $K = 5$. Figure 2 shows some example scenarios comparing trajectory deviations induced by the attack patterns discovered by the two algorithms.

Recall that GradOpt trains with frames collected from four trajectories as a means of approximately accounting for self-correcting behavior of the control network, as it begins to deviate from its original trajectory which would result in frames from different viewpoints. To evaluate the importance of this approach, Table 1 also includes an ablation measuring average deviation when optimizing with GradOpt using only one ($T = 1$) trajectory. We find that the discovered attack patterns in this case are less successful when used in actual simulation, leading to lower average deviation scores than optimizing with multiple trajectories—although these scores remain competitive with those from BO.

| K #Rect. | Straight | | | Left | | | Right | | |
|---|---|---|---|---|---|---|---|---|---|
| | BO | GradOpt | % ≥ | BO | GradOpt | % ≥ | BO | GradOpt | % ≥ |
| 1-b | 2.23 | 1.60 | 88% | 4.70 | 4.75 | 92% | 5.14 | 6.39 | 83% |
| 1 | 2.01 | 2.79 | 81% | 5.53 | 5.25 | 60% | 6.11 | 8.42 | 83% |
| 2 | 1.89 | 4.41 | 100% | 5.28 | 5.60 | 67% | 7.20 | 9.85 | 67% |
| 3 | 2.73 | 4.30 | 88% | 5.47 | 5.53 | 75% | 5.98 | 8.82 | 92% |
| 4 | 0.91 | 4.11 | 100% | 5.00 | 7.00 | 83% | 6.07 | 8.86 | 83% |
| 5 | 1.66 | 6.20 | 94% | 5.60 | 7.05 | 70% | 7.83 | 9.46 | 83% |

Table 2: Infraction penalties for each type of scenario where the expected trajectory is driving straight, left, or right. These results are for simulations in "clear noon" conditions.

| K #Rect. | Cloudy Noon | | | Rainy Noon | | | Clear Sunset | | |
|---|---|---|---|---|---|---|---|---|---|
| | BO | GradOpt | % ≥ | BO | GradOpt | % ≥ | BO | GradOpt | % ≥ |
| 1-b | 2.29 | 3.89 | 85% | 2.69 | 3.87 | 82% | 2.41 | 2.60 | 88% |
| 1 | 3.19 | 3.28 | 82% | 2.98 | 3.33 | 82% | 3.36 | 3.28 | 90% |
| 2 | 2.87 | 5.29 | 90% | 4.03 | 5.03 | 80% | 2.92 | 3.75 | 75% |
| 3 | 3.06 | 4.85 | 88% | 3.28 | 5.61 | 88% | 2.60 | 3.60 | 93% |
| 4 | 2.39 | 5.80 | 88% | 3.18 | 5.67 | 78% | 2.38 | 4.41 | 93% |
| 5 | 3.56 | 5.50 | 85% | 3.81 | 5.53 | 80% | 2.67 | 5.65 | 90% |

Table 3: Infraction penalties, across all scenarios, in simulations with weather conditions different from the one used for attack optimization ("clear noon").

We take a closer look at the vulnerability of different types of scenarios by separately reporting infraction penalties for each in Table 2. We see that scenarios where the expected behavior is driving straight are the hardest to attack, likely because they are the simplest to drive in. BO tends to achieve only a moderate infraction score in these settings, even at higher values of $K$. In contrast, GradOpt reveals that even these scenarios are in fact vulnerable when one is allowed to consider more complex attack patterns—achieving an average deviation that is four times higher than BO at $K = 5$. Conversely, driving right scenarios are the most vulnerable with both methods being successful even with simple patterns, with GradOpt again yielding higher deviation and more infractions.

Finally, we evaluate the robustness of our attack approach by evaluating their success in different climate and visibility conditions than those used for attack optimization. Table 3 presents results for simulations with four such climate settings, and as expected, we find that both BO and GradOpt do see a drop in penalty scores compared to the standard setting in Table 1. Nevertheless, most of the attacks induce infractions, especially at higher values of $K$, with GradOpt again being significantly more successful than BO.

## 4 CONCLUSION

A great deal of attention has been devoted to understanding adversarial perturbations in computational perception, with autonomous driving the most common motivation. However, most prior research has considered these to be independent for each input image. In contrast, autonomous driving is dynamic, and even if a perturbation succeeds to fool the car in a particular frame, it can fail in another frame, and he car can self-correct. Thus, to fully understand vulnerabilities in autonomous vehicle architectures we need to evaluate them in an end-to-end fashion using driving scenarios. However, this is inherently challenging because the resulting experimentation, be it using a simulator or actual driving, is extraordinarily time consuming. We developed a novel framework that allows us to largely avoid costly driving experiments, relying instead on a novel compositing approach which is fully differentiable. Our approach is significantly more potent at discovering physically realizable adversarial examples than prior art, while also requiring far fewer runs of actual driving trials or simulations. Moreover, we find that the vulnerabilities we discover are robust, and persist even under variability in weather conditions.

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

APPENDIX

In this appendix, we provide further details of our shape parameterization and calibration, as well as additional results and visualizations from our experiments.

## A    PATTERN SHAPE PARAMETERS

Each rectangle in our shape is parameterized by four values $x_k^S = [w, l, o, \theta]$, corresponding to width, length, horizontal offset, and rotation or orientation. These parameters are defined with respect to the top-down view of a $400 \times 400$ pixel "canvas" that is composited onto the road surface. Each rectangle is first drawn aligned with the $x-$ and $y-$axes of this canvas to be of width $w$ and length $l$ pixels, and vertically centered and horizontally offset so that its left edge is at $x = o$, and then rotated about the center of the canvas by angle $\theta$. Portions of rectangles that lay outside the canvas after this process were clipped from the pattern. Our parameterization expands on the one originally used by Boloor et al. (2020) in two respects: by searching over length $l$ instead of fixing it to the length of the canvas, and having a separate orientation $\theta$ for each rectangle rather than a common one for all rectangles.

## B    CALIBRATION DETAILS

We estimate homographies between the canvas and each frame from 60 corresponding pairs as described in Sec. 3, using a direct linear transform. While doing so, we ensure the matrix has the correct sign so that homogeneous co-ordinates of points projected in the frame have a positive third co-ordinate when they are visible, and a negative one when they are "behind the camera". When compositing patterns on the frame, this allows us to retain only the portion of the pattern that would be visible from that viewpoint. The color transforms are estimated simply from the color correspondences using a least-squares fit.

## C    ADDITIONAL RESULTS

In our main evaluation, we reported deviations (without cars or pedestrian) only in the overall evaluation in Table 1, and reported infraction penalties *with* cars and pedestrians for all other comparisons. For completeness, we report deviation scores for those comparisons here, as well as infraction penalties computed in simulations without cars or pedestrians (with the caveat that some of the highest penalties defined by the challenge are for collisions with pedestrians and cars).

Table 4 reports deviation scores separately for different types of scenarios, and Table 5 the corresponding infraction penalties when driving without pedestrians or cars. Finally, in Tables 6-8 we report both deviation and car and pedestrian-free infraction penalty scores for simulations in the different non-standard weather conditions. We find that these results are qualitatively consistent with those in our main evaluation in Sec. 3.

| K | **Straight** | | | | **Left** | | | | **Right** | | |
|---|---|---|---|---|---|---|---|---|---|---|---|
| #Rect. | BO | GradOpt | $\% \geq$ | BO | GradOpt | $\% \geq$ | BO | GradOpt | $\% \geq$ | | |
| 1-b | *0.50* | 0.56 | *62%* | 0.93 | *1.43* | *92%* | 1.04 | *1.12* | *67%* | | |
| 1 | 0.48 | *0.61* | *69%* | 0.91 | *1.39* | *83%* | 1.30 | *1.13* | *67%* | | |
| 2 | 0.49 | *0.82* | *88%* | 1.10 | *1.47* | *92%* | 1.09 | *1.30* | *67%* | | |
| 3 | 0.52 | *0.81* | *69%* | 1.27 | *1.59* | *83%* | 1.04 | *1.42* | *75%* | | |
| 4 | 0.42 | *0.86* | *75%* | 0.93 | *1.82* | *100%* | 0.98 | *1.34* | *67%* | | |
| 5 | 0.45 | *0.97* | *81%* | 0.96 | *1.68* | *92%* | 1.15 | *1.44* | *75%* | | |

Table 4: Deviations in simulations without cars or pedestrians, in standard "clear noon" weather conditions, for each type of scenario.

| K #Rect. | BO | Straight GradOpt | % ≥ | BO | Left GradOpt | % ≥ | BO | Right GradOpt | % ≥ |
|---|---|---|---|---|---|---|---|---|---|
| 1-b | *1.62* | 1.00 | *88%* | 2.33 | *3.75* | *83%* | 3.33 | *4.08* | *67%* |
| 1 | 1.50 | *1.88* | *75%* | 4.17 | *4.25* | *75%* | 5.17 | *7.33* | *83%* |
| 2 | 0.75 | *1.88* | *100%* | 2.67 | *4.17* | *83%* | 6.00 | *6.50* | *67%* |
| 3 | 2.12 | *3.12* | *81%* | 3.67 | *4.17* | *75%* | 5.17 | *6.33* | *92%* |
| 4 | 0.50 | *2.38* | *94%* | 1.67 | *4.58* | *92%* | 5.33 | *7.42* | *92%* |
| 5 | 1.62 | *3.62* | *88%* | 3.00 | *5.25* | *75%* | 6.33 | *8.33* | *100%* |

Table 5: Infraction penalties *without cars or pedestrians* for each type of scenario, in standard "clear noon" simulations. Note these differ from our standard infraction penalties in Sec. 3 where penalties are computed *with* simulated cars and pedestrians.

| K #Rect. | BO | Deviation GradOpt | % ≥ | BO | Infraction Penalty GradOpt | % ≥ |
|---|---|---|---|---|---|---|
| 1-b | 0.90 | *0.90* | *90%* | 1.05 | *2.40* | *72%* |
| 1 | 0.51 | *0.97* | *90%* | 1.65 | *2.45* | *80%* |
| 2 | 0.59 | *1.16* | *95%* | 2.00 | *3.65* | *85%* |
| 3 | 0.73 | *1.17* | *97%* | 1.10 | *3.35* | *90%* |
| 4 | 0.74 | *1.22* | *95%* | 1.30 | *4.20* | *82%* |
| 5 | 0.56 | *1.30* | *90%* | 1.90 | *4.35* | *88%* |

Table 6: Deviation and infraction penalties, *both* computed without cars or pedestrians, over all scenarios in "cloudy noon" conditions.

| K #Rect. | BO | Deviation GradOpt | % ≥ | BO | Infraction Penalty GradOpt | % ≥ |
|---|---|---|---|---|---|---|
| 1-b | 0.77 | *1.0* | *75%* | 1.05 | *2.30* | *90%* |
| 1 | 0.56 | *0.96* | *78%* | 1.50 | *2.95* | *85%* |
| 2 | 0.47 | *1.05* | *88%* | 1.60 | *3.15* | *93%* |
| 3 | 0.83 | *1.07* | *80%* | 2.10 | *3.80* | *85%* |
| 4 | 0.84 | *1.13* | *80%* | 1.60 | *3.90* | *88%* |
| 5 | 0.47 | *1.20* | *88%* | 1.60 | *3.95* | *97%* |

Table 7: Deviation and infraction penalties, *both* computed without cars or pedestrians, over all scenarios in "rainy noon" conditions.

| K #Rect. | BO | Deviation GradOpt | % ≥ | BO | Infraction Penalty GradOpt | % ≥ |
|---|---|---|---|---|---|---|
| 1-b | 0.50 | *0.73* | *65%* | 1.30 | *1.25* | *65%* |
| 1 | 0.40 | *0.70* | *75%* | 1.55 | *2.00* | *75%* |
| 2 | 0.40 | *0.90* | *90%* | 1.05 | *1.95* | *90%* |
| 3 | 0.39 | *0.94* | *85%* | 0.75 | *2.40* | *85%* |
| 4 | 0.40 | *1.10* | *93%* | 1.45 | *2.80* | *93%* |
| 5 | 0.37 | *1.19* | *90%* | 1.70 | *2.60* | *90%* |

Table 8: Deviation and infraction penalties, *both* computed without cars or pedestrians, over all scenarios in "clear sunset" conditions.

# D    VISUALIZATION

Finally, we use an example "drive straight" scenario to visualize the behavior of the controller when driving with attack patterns, with $K = 4$ rectangles each, returned by GradOpt and BO. We show these patterns, in the top-down canvas view, in Fig. 3. Then, we show frames from the vehicle's camera feed as it drives through the road with the respective patterns painted on the road in various climate conditions, in simulations with pedestrians and cars in Fig. 4, and without in Fig. 5.

For this scenario, the pattern returned by BO is unable to cause a significant deviation in the vehicle's trajectory as it drives across the stretch of road with the pattern painted on it. In contrast, GradOpt's pattern is able to cause the car to veer sharply to the left in all but the "clear sunset" climate setting (which as we see looks visually the most different from the base setting)—causing it to crash into another car in Fig. 4 and the opposite sidewalk in Fig. 5.

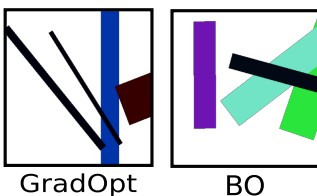

GradOpt          BO

Figure 3: Attack patterns with $K = 4$ rectangles returned by GradOpt and BO for the example "drive straight" scenario illustrated in Figures 4 and 5.

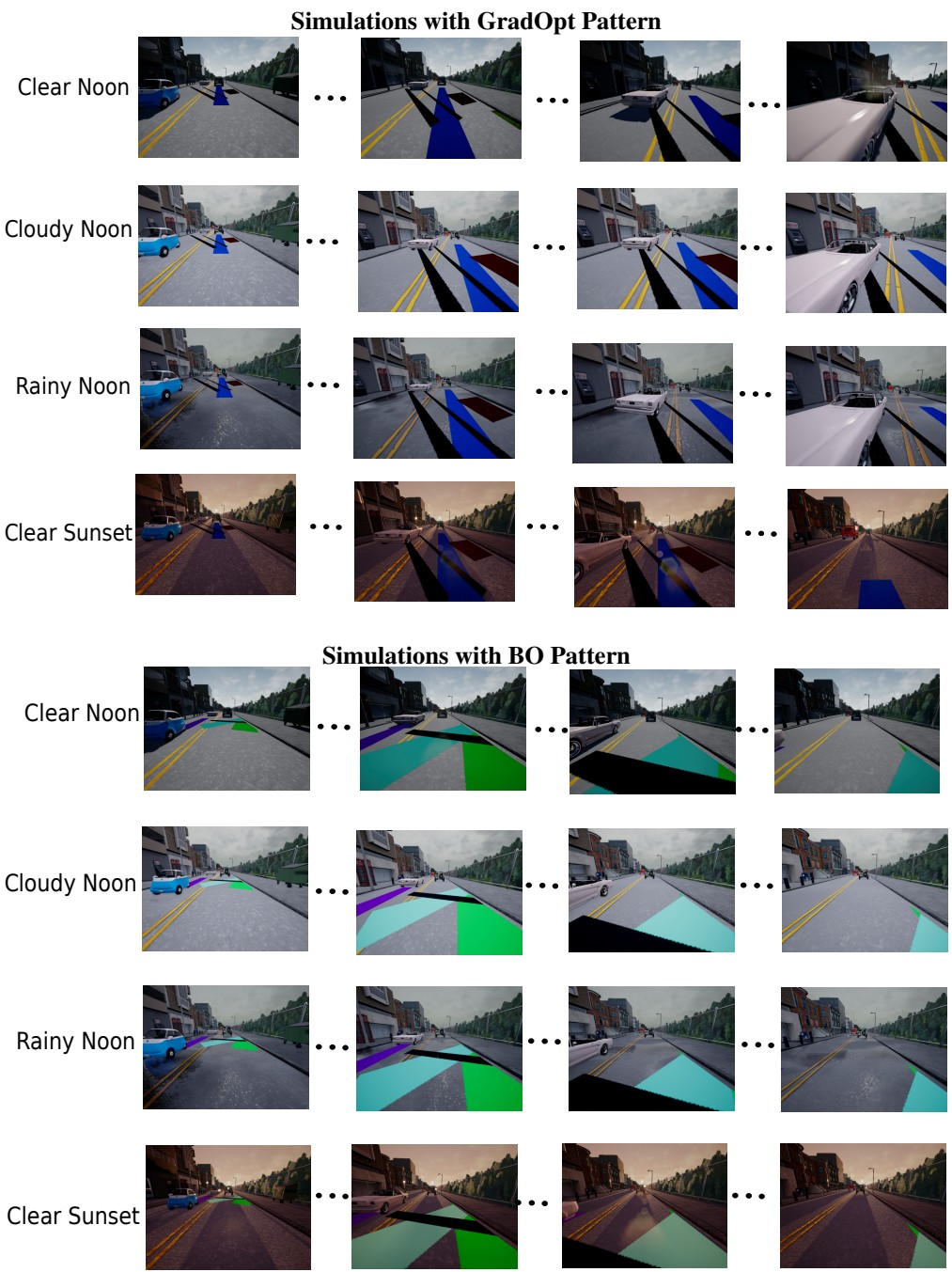

Figure 4: Frames from driving simulations, with cars and pedestrians in different weather conditions, after introducing attack patterns from GradOpt (top) and BO (bottom).

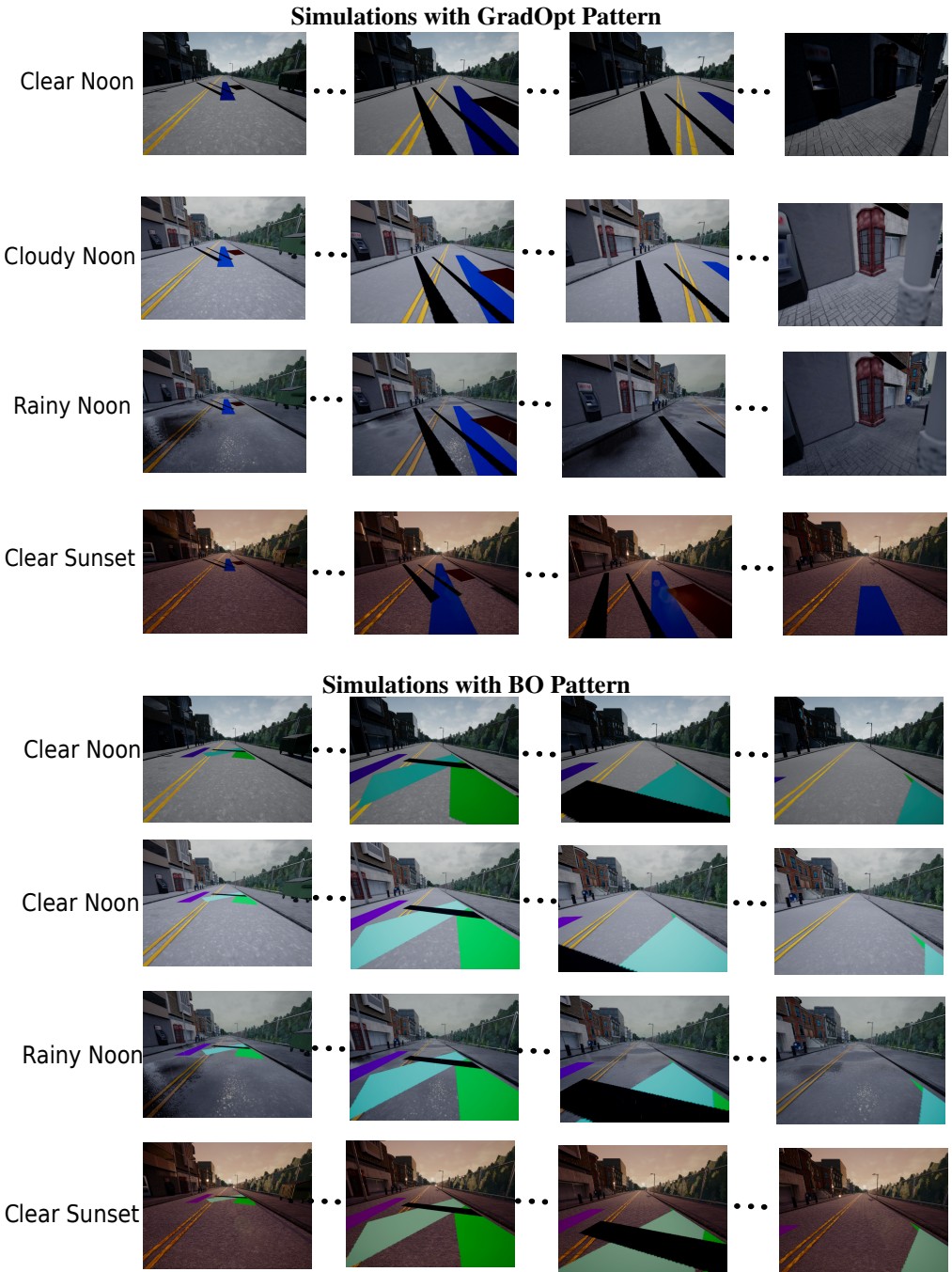

Figure 5: Frames from driving simulations, without cars or pedestrians in different weather conditions, after introducing attack patterns from GradOpt (top) and BO (bottom).

