# OpenReview forum: "Finding Physical Adversarial Examples for Autonomous Driving with Fast and Differentiable Image Compositing"
_ICLR.cc/2021/Conference — Reject_

### Official Review · AnonReviewer1 · 2020-10-27
**A more efficient approach for finding physical adversarial examples in autonomous driving simulation**

**Rating:** 6
**Confidence:** 4

**Review:**

The paper proposes an end-to-end differentiable method for finding adversarial patterns to be added to the environment for autonomous driving. It utilizes image composition with homography thus it can compose the adversarial pattern into the image frames of all image frames of a driving sequence. Combined with a neural-network based controller which outputs the steering angle, the proposed method can find adversarial examples more efficiently comparing to a Bayesian optimization(BO) based baseline while resulting in trajectories with greater deviation.

The proposed method relies on an approximation of the image frames of the trajectories by adding random noise to the controller outputs and use those trajectories for learning. This dramatically reduces the number of calls to the simulator comparing to the baseline BO method. The paper states that "Given the fact that actual control is closed-loop, it is not evident that this simple approach would work; however, our experiments below show that it is remarkably effective, despite its simplicity." Is it possible the reason for this approximation to work is that the scenarios are not complicated enough? If for more complicated scenarios where more calls to the simulator are needed, the benefit of the proposed method over the baseline would be much smaller.
- Related question: are the trajectories with the noisy control resulting in infraction or not?

Other questions / comments:
1. The choice of using four trajectories as an approximation seems random. Any ablation study on the number of trajectories?
2. The results in the three tables only contain average values without standard deviation.
3. In sec 2.3 "we use the absolute value of the sum of the differences between the angles as our divergence metric." It seems to me that the sum of the absolute values of the differences between the angles makes more sense because the positive and negative difference values would cancel out without taking absolute values before the summation. Similarly in Sec 3.1.
4. Related to 3, I'm also curious about the design choice of using the controller deviation as the optimization objective function while using the trajectory deviation to measure the effectiveness of the adversarial attack. Those two may not necessarily correlate (An example will be if the car is supposed to go straight, while zero steering and both a left and a following correcting right steering will keep the car straight thus resulting in very similar trajectory, the controls are more different. On the other hand, a single left or right steering will result in large trajectory error as it accumulates.)
5. All experiment results are in simulation so it's hard to draw conclusion regarding real driving scenarios.

---

> ### Author Response · Authors · 2020-11-17
> **Author response to reviewer 1**
>
> - The main benefit of our "trajectory data augmentation" approach is that it allows us to train our attack to be effective when observed from a diversity of viewpoints. We evaluate this on a number of intersections and scenarios in the Carla town. We believe that this strategy will be successful as long as deviations across a small stretch of road are enough to cause an infraction/crash (because then our trajectories will sample a representative enough set of diverse viewpoints). But that is a constraint anyway since we allow ourselves to only modify a small stretch of road. For attacks that must be realized over a longer horizon, you are right: that would probably require a closed loop.
>
> - The noisy trajectories do indeed result in infractions.
>
> - Our choice of four trajectories was to balance the need for multiple views with the need to minimize the number of calibration simulator calls. We’ve already shown an ablation when using only one trajectory in Table 1, and will add ablations for results with 2 and 3 trajectories to the camera ready (essentially, effectiveness increases with more trajectories).
>
> - Since the possible infraction scores can change from scenario to scenario, rather than showing standard deviations, we opted for a "sign-test" as showing statistical significance. All our tables include the %>= metric, which shows that our approach leads to a higher infraction (or deviation) than BayesOpt in a large fraction of scenarios and settings.
>
> - While our evaluation is in simulation (it would be impractical/dangerous to do this on real cars), we evaluate generalizability by looking at how well our attacks transfer over to different visibility and weather conditions (Table 3), where the video frames are quite different from those used for optimization (see Figs 4 and 5 in the appendix). This shows that the attacks our method finds are "robust", and would transfer over to real scenarios.

---

### Official Review · AnonReviewer4 · 2020-10-28
**Interesting setup while less practical for real scenarios**

**Rating:** 6
**Confidence:** 4

**Review:**

To summarize, the authors propose a road-painting attack with rectangles to deceive a controller network such that the car will deviate from the correct trajectory. The simulation is done on CARLA.

**The threat model**
Painting roads with rectangles is very interesting. The closest one I saw is patching stop signs with rectangle markers [Eykholt 2018] as cited in the paper. Meanwhile, this setup brings many questions.

First, the attack space is incredibly small (parameterized color rectangles on the road). With such a small space, I would expect the space allowed for changing the controller network output is also small. On the contrary, in traditional pixelwise adversarial attack, the attack vector is high-dimensional so the vulnerabilities can stack up to change the network output to arbitrary values. In the current setup, such stacking-up is unlikely to happen. Therefore, even the attack is successful, I believe tuning with gradient-based or gradient-free optimizers does no help much; the baseline attack success rate could be already high.
In other words, I doubt that even random rectangles may already cause deviations and infractions. Searching with BayesOpt or GradOpt may not help much; there could be a wide range of parameters that can cause reasonable deviations already.

This setup is also perceivable by humans and thus not stealth. In traditional adversarial attacks, the perturbations are small enough to be ignored by humans but will cause a deep network to fail; the current setup is not the case; therefore it can be easily defended by humans.

**GradOpt v.s. BO**
- BO is a black-box optimizer that has no access to the inner structure of the controller. GradOpt in this paper is a white-box model and it is unfair to compare BO with GradOpt.
- GradOpt is a very standard gradient-based optimization pipeline for adversarial attacks involved with 3D projection and rendering.  Judging from the images, I believe the rectangles are not shaded, so an affine transformation already suffices. This limits the novelty of the method.

**Constraints for the attack vector**
I could not find the description of the constraints for the attack vectors. If there is no constraint; then there exists a trivial solution: just paint the road to a constant color using an infinitely large rectangle and there are no lanes anymore. It will be very interesting to see the evaluation metrics in this case just for ablation purposes.

**Transferability**
With a small attacking space, I am curious whether GradOpt learns to find the vulnerabilities in the network, or learns to cover the important regions on the road. Also it is unclear whether it will succeed for real images and videos. The submission lacks transferability experiments to study those scenarios.

[Update after reading authors' comments]
The critical issue is resolved: it seems their method is indeed better than plain BO, which in turn is better than random parameters. Though I still have a little doubt about how practical it can be in real scenarios, I increase the score to marginally accept.

---

> ### Author Response · Authors · 2020-11-17
> **Author response to reviewer 4**
>
> - Our attack space is chosen following the Boloor et al. (BayesOpt) paper and is motivated by the fact that such a pattern is more realistic to implement physically (i.e., paint on roads). Indeed, this makes optimization harder than for say a per-pixel pattern attack (a modified version of our method could fix all the rectangles were small squares that formed a grid and estimate just colors for each square for which we could compute gradients directly: but this would perhaps not be physically realizable).
>
> - The random rectangles question is an interesting one, and actually the BayesOpt paper did that precise study: they demonstrated that randomly sampling parameters (even in the low dimensional parameter space they considered) was unable to find a successful attack. We apologize for not discussing these results in our paper and will do so in the camera ready. Moreover, note that as our results show, we are able to find successful attacks very frequently when BayesOpt can not: which implies that the optimization is non-trivial.
>
> - Vs BO: Note that while BO is a black-box method, the approach requires calling not just the controller, but also a physics engine (or physically driving the car), many times. Since we evaluate our method and BO with the same number of optimization iterations, we have the same number of calls to the controller, where we also additionally have access to gradients. But note that we do not need to call the physics engine as many times.
>
> Note that the main novelty of our work is that it is the first to realize a gradient-based attack on the closed loop driving controller, by using an approximate rendering approach that is shown to successfully transfer over to the full physics-based simulator.
>
> - Stealth/Constraints: All of our patterns are constrained to lie on a small patch of road, i.e., within our 400x400 canvas in the reference view (appendix A). Therefore, while the patterns are obvious when seen, they are only visible when one is close to the intersection (in that sense, it is similar to the sticker attack). We will add visualizations of the canvas area in the camera ready.
>
> - Covering the entire canvas (not the entire road) with a uniform color rectangle does not lead to a successful attack. This suggests that our optimizer is actually finding patterns that confuse the controller about the layout of the road.
>
> - Transferability: We are not able to evaluate these patterns on a real car (this would be impractical and dangerous). But we do run transferability experiments to other visibility and weather conditions (see Table 3) than what was used for optimization. This change in conditions leads to a significant change in visual appearance, as seen in Figures 4 and 5 in the appendix.

---

### Official Review · AnonReviewer2 · 2020-10-28
**A scalable framework for generating effective adversarial examples for driving scene**

**Rating:** 5
**Confidence:** 4

**Review:**

This paper proposes a scalable and efficient approach for finding adversarial physical modification to the video inputs of autonomous driving. Assuming the perturbations are in form of a collection of several rectangles, the model parameterizes the physical modifications. By simply ignoring the closed-loop of viewpoint sequence and frames, the model directly creating adversarial frames with compositing methods. Some approximated algorithms are used to ensure the model can be optimized by the gradient-based method. With the improvement above, the iteration speed of the model is greatly improved.

Strengths:
+ This paper proposes a highly scalable framework for designing physically realizable adversarial examples against end-to-end autonomous driving architectures, which makes the much stronger attack results.
+ In the simulated climate settings, the proposed method demonstrates robustness against unforeseen variations in weather and visibility.
+ With a small number of initial calibration running, the search for optimal parameters can be carried out with end-to-end gradient-based optimization, instead of relatively slower Bayesian Optimization.

Weaknesses:
+ It lacks of a convincing explanation about why ignoring the closed-loop of viewpoint sequence and frames the model can still work (in Section 2.2).
+ The mask function is still not differentiable in most cases, thus approximation should be adopted. The authors should clarify how much such approximation affects performance.
+ There are too many long sentences makes some parts of the article difficult to understand. Please try to improve the presentation

I hope to hear the authors response regarding the weakness listed above during the rebuttal period.

---

> ### Author Response · Authors · 2020-11-17
> **Author response to reviewer 2**
>
> - We will expand on the intuition behind why our approach works despite the lack of a closed loop during optimization. Essentially, by considering different random trajectories, we're ensuring that our pattern is "confusing" to the controller from a diversity of viewpoints (in that, it causes a high deviation from the ideal control signal without the pattern). It is because we train on this diverse training set of viewpoints that we are able to generalize to the viewpoints the controller actually encounters during the simulation. Essentially, this is a data augmentation approach.
>
> - The approximation we use is similar to the one used in Spatial transformer networks, which has been used successfully in other application settings. Note that we only approximate this in the backward pass for backprop, not when computing the mask itself in the forward pass. The fact that the approximation works well is demonstrated by the success of our optimization strategy in discovering optimal attack patterns in many different scenarios and with a large number of parameters.
>
> - We appreciate the feedback on the presentation and will update the final camera ready to improve readability.

---

### Official Review · AnonReviewer3 · 2020-11-02
**motivation not strong enough**

**Rating:** 5
**Confidence:** 2

**Review:**

This paper presents an approach to design physical adversaries to attack end-to-end autonomous driving systems. The proposed approach maps adversarial patterns onto video frames recorded from real world to generate adversarial examples for deviating the control of a vehicle.

The problem considered by this work is significant as physical adversarial attacks pose serious threats to the safety of autonomous driving. The paper is well written and most of the technical details are clearly articulated. The experiments demonstrates the efficacy of the proposed solution with results reported under different physical conditions .

One main concern I have about this work is that its motivation does not seem strong enough. Since both the training and evaluation are performed based on simulation, how real video streams help in developing a more robust approach remains unclear to me. The authors may need to more clearly argue why simulated images are less advantageous in this case.

The geometric and color parameters are obtained from manual calibration in this work. While this makes learning easier, designing an automatic matching method to learn these parameters from data would be more interesting, and potentially leads to more robust attacks against realistic environments.

Some other comments:
1) It’s a little bit difficult to understand why a few painted boxes on the road could fool an autonomous system. It would be helpful if the authors could provide some explanations of how these physical modifications affect the vehicle's controller.
2) How are M and V determined in Eq. (1)?

---

> ### Author Response · Authors · 2020-11-17
> **Author response to reviewer 3**
>
>
> - While our evaluation is in a physics engine simulator, our attack optimizer is far less reliant on it than the existing BO approach. In particular, we need to only run the simulator a small number of times to get our reference frames and for calibration, but then the entire optimization happens using our "approximate" rendering by composition. This has two implications:
>
> a) This means that it makes it more likely our approach can be transferred to an actual vehicle: we drive it a few times along a stretch of road to collect real frames and calibrate, carry out the optimization offline, and then return and paint the road with the actual pattern. While we don't actually evaluate on a real vehicle (this is impractical but more importantly dangerous), we argue that an attack strategy that doesn't require an actual physical loop (in reality or simulation) is more potent.
>
> b) Moreover, even in the setting where everything is in simulation, we need to call the "real" physics-engine simulator far fewer times (74 instead of BO's 1000). Since simulations are computationally very expensive, this gives our approach a significant advantage.
>
> - Note that although we discuss manual calibration as an option, we actually computed the calibration patterns automatically in our experiments as discussed in Sec. 3: we drew calibration patterns on the road and automatically detected these in the frames. Some form of calibration is unavoidable in our setting, since we need to know how to transfer our computed attack pattern back to the physical environment.
>
> - Neural network-based systems are known to be vulnerable to adversarial patterns that would not confuse humans. Our work just finds these patterns under the restrictions that they correspond to modifications of the physical environment, rather than digital manipulation of images.
>
> - M is determined by the choice of shape (rectangles in our case as discussed later). V are just the masks applied in a predetermined order signifying which shape is "on top".

---

### Decision · Program_Chairs · 2021-01-07
**Final Decision**

**Decision:**

Reject

**Comment:**

Thank you for your submission to ICLR.  Overall the reviewers and I think that this paper presents some nice contributions to the adversarial attacks literature, demonstrating a low-sample-complexity, "physically-realizable" attack in a domain of clear importance and interest in machine learning.  The move to considering more "in the loop" adversarial examples is particularly compelling, and the threat model and improvement over BO methods are both compelling here.

The main downside of this paper, of course, is the fact that the "physical adversarial examples" are of course nothing of the sort: they are simulated.  Rather, they are just simulated in a manner that may plausibly be slightly more amenable to real-world deployment. The authors claim that they don't carry out an evaluation on a real system because it is "dangerous" is a bit overly dramatic: the tests could easily be carried out in a controlled environment, and demonstration on an actual physical system (even, e.g., and RC car) would vastly improve the impact of this work.  As it is, the paper is borderline, but ultimately slightly below the high bar set by ICLR publications.  I would strongly encourage the authors to reconsider the inclusion of the word "physical" in the title, as it honestly sets expectations high for a promise that the paper cannot deliver on, or (even better) to run real experiments on even a small physical system, demonstrating the transferability there.  The paper ultimately has the potential for a high impact in this field, if these issues are addressed.